# The influence of college students' aesthetic cognitions on aesthetic behaviours: The Chain mediation effect

Qiao Qiao, Yongzhi Jiang ⬤ *

Department of Educational Management, International College, Krirk University, Bangkok, Krung Thep Maha Nakhon, Thailand

* psy_yongzhi@126.com

**Data Availability Statement:** All relevant data are within the paper and its Supporting Information files.

**Funding:** The author(s) received no specific funding for this work.

## Abstract

This research investigates the psychological and behavioural mechanisms of college students' aesthetic behaviours. A survey was administered to 1,060 students attending general undergraduate universities and measured four structured scales: aesthetic cognition, aesthetic emotion, aesthetic value tendency and aesthetic behaviour. The responses were scored with a 5-point Likert scale. Structural equation modelling was used to construct the measurement model and structural model. The survey results indicate a positive correlation among the four variables. Second, aesthetic cognition has a direct and positive predictive effect on college students' daily aesthetic practices. Furthermore, aesthetic cognition influences aesthetic behaviour through the chain mediating effect of aesthetic feeling and aesthetic preference. Accordingly, we conclude that the concrete path and mechanism of college students' aesthetic cognitions affect their aesthetic behaviours. Specifically, college students' aesthetic cognitive abilities are conducive to their cognitions of positive aesthetic feelings and their cultivation of aesthetic standards, boosting the development of their daily aesthetic practice.

## 1 Introduction

In recent years, the development of aesthetic education has received a great deal of attention in Chinese education. Aesthetic education aims to improve the aesthetic qualities of Chinese university students and has positive significance in cultivating noble aesthetic pursuits and noble personalities among university students. Aesthetic behaviour is the outwards manifestation of aesthetic literacy, aesthetic common sense and aesthetic concepts, which constitute the aesthetic qualities of individuals; these can be concretely expressed only through aesthetic behaviour [1]. Aesthetic education evokes and forms characteristics and attributes with human value in students [2]. However, some scholars claim that aesthetic education is not effective [3–6]. Shi O and Hou Jingmin [7] suggest that there is a substantial difference between passive viewing and active participation in the creation of beauty for improving aesthetic literacy, that students' aesthetic literacy is not only reflected in their artistic activities, that aesthetics should be

**Competing interests:** The authors have declared that no competing interests exist.

infused into life, that the act of aesthetics is a form of aesthetic creativity, and that this ability to create beauty is expressed in the ability to create and express using diverse materials and multiple methods [8]. This study posits that the aesthetic practice behaviours of university students can be divided into three aspects: the behavioural habit of participating in art, the beautification of the living environment, and the rational matching of clothing, moreover, the three aspects represent the fit and communication between the aesthetic subject and object, which comprehensively reflects the quality of human existence and is the most typical mass aesthetic activity. This involves a kind of creation in which the relationship between the subject and object is reciprocal and constitutes the action of the subject in consciously interacting with the aesthetic object to achieve pleasure and creation in life [9]. There is very little research on the everyday aesthetic behaviour of university students. This study uses structural equation modelling to conduct an empirical study. The study of the psycho-behavioural mechanisms of the aesthetic behaviours of university students is important for understanding the characteristics of aesthetic psychology and ensuring high-quality development of aesthetic education.

The development of aesthetic cognition plays an important role in the formation and development of the entire aesthetic psyche. According to cognitive-emotional theory, the influence of aesthetic cognitive processes on aesthetic experience ranges from low-level aesthetic perception to high-level aesthetic understanding and evaluation, with the level of aesthetic cognition determining the level of aesthetic experience. Multilevel theory suggests that aesthetic experience can occur without conscious aesthetic cognition. Aesthetic cognition, then, is the basis of aesthetic experience, and aesthetic emotions are formed by individuals during any aesthetic process by in turn responding to aesthetic cognitive processes [10,11]. According to Zhang Dayun, aesthetic cognition is aesthetic information processing; it involves the input, encoding, transformation, storage and extraction of aesthetic information [12]. Research by neuroscientists within the field of aesthetic cognition indicates that experts trained in specialised knowledge and artistic skills, or artists, have above-average perceptions, equipping them with well-established perceptual patterns and stronger aesthetic cognitive abilities [13]. Zeki also argues that artists use artistic techniques to explore the principles of visual processing in the brain and unconsciously apply the principles of the brain's processing of various visual attributes in the act of creation [14]. These professionals are more adept in using aesthetic knowledge, more effective in visual analysis and formal cognition and thus in creating things of beauty.

Aesthetic emotions are pleasurable experiences or emotional reactions to beauty that accompany the appreciation of aesthetic objects. "Aesthetic feeling" is defined in psychology as a complex emotion unique to human beings. Aesthetic feeling is a sublime and beautiful emotion that people experience when they appreciate natural things and works of literature and art. It is the experience, the pleasurable feeling that arises when people appreciate beautiful natural objects, works of art and other human products [15]. Beauty is a feeling of pleasure triggered by the qualities of an aesthetic object itself [16], the emotion of pleasure that a subject experiences after the aesthetic cognitive processing of a meaningful form [17,18].

Aesthetic interest is an experience of aesthetic pleasure that arises in the controlled processing stage, where the aesthetic object is finely processed [19]. Temme has explored the impact of an individual's aesthetic knowledge on his or her aesthetic experience; the amount of information an individual has about an artwork thus affects his or her aesthetic experience of beauty in a museum setting, whereby an individual's ability to have a more aesthetically pleasing experience requires possessing background information about a given artist [20]. Millis has demonstrated that title information can impact aesthetic evaluation [21]. Russell also finds that people's evaluations of the pleasantness of their work increases with the interpretation of the meaning of that work [22]. Additionally, experimental results among scholars such as Leder suggest that subjects' knowledge of a work can enhance their positive emotions [23]. For

instance, Lengger finds that individuals' knowledge of stylistic artwork information can improve their perceptions of paintings and that their knowledge of artwork style can enhance their categorisation of works for smoother aesthetic cognitive processing [24].

Thus, this study's first hypothesis is as follows:

H1: The aesthetic cognitions of university students significantly and positively influence their aesthetic emotions.

Graf and Landwehr have therefore proposed a dual processing model of fluency, the pleasure-interest aesthetic (PIA) model, based on the fluency model of information processing. According to the PIA model, the experience of aesthetic pleasure consists of aesthetic pleasure in the automatic processing stage and aesthetic interest in the controlled processing stage of the aesthetic process. In the primary stage, the individual perceives the pleasurable experience of an aesthetic object without the involvement of other media; it is active and spontaneous. In the advanced stage of the aesthetic process, the individual devotes attentional resources to the aesthetic object and finely processes it to experience aesthetic pleasure. A successful aesthetic cognitive process thus activates the reward system and thereby triggers aesthetic pleasure [25] via the activation of the brain's reward centre for artwork [26]. Aesthetic processes, then, are pleasurable [27], and aesthetic needs and aesthetic cognitive rewards—positive emotions—are the driving forces in aesthetic behaviour.

The extended-constructive theory of positive emotions suggests that positive emotions expand an individual's cognitive range, increase one's number of cognitive resources for making connections, distract one's attention from more complex situations, and increase the breadth of one's relevant resources for problem solving. This leads to more effective information being available to solve problems. By expanding the range of available resources, the behavioural guidance system becomes altered, and individuals become motivated to seek new and creative ways of thinking and acting, producing a positive "emotion-cognition-behaviour" spiral [28]. Indeed, empirical research has shown that positive emotions, whether spontaneous or induced, lead to greater creativity and flexibility in thinking and acting [29] and that positive emotions have a more significant effect on creativity than negative or neutral emotions [30].

Attribution theory also suggests that people with positive emotions are more likely to attribute their failures to their external environment [31]. Such attributions may help individuals maintain a higher level of creative efficacy and thus feel more confident in their subsequent creative work and become better able to reverse previous creative failures.

Thus, this study's second hypothesis is as follows:

H2: The aesthetic emotions of university students significantly and positively influence their aesthetic behaviours.

Aesthetic values have become one of the most general and pervasive values in the value system of life [32]. Similarly, aesthetic needs, from the perspective of aesthetic psychology, are both instinctive and represent a higher need in terms of the origin of the adornment and aesthetic values of clothing. The humanist psychologist Maslow has referred to aesthetic needs as developmental, transcendental and postexistential, which are expressed in the need for structure, order, regularity and symmetry [33]. Beneath these aesthetic needs lies a higher level of human need—to realise oneself, to gain a social identity and self-identity, to show one's personality and to express one's aesthetic and artistic potential. Aesthetic needs at the real level and aesthetic needs at this higher level advance each other and combine with more external realities to compose the external form of one's aesthetic tendencies, which is a certain kind of interest.

A deeper psychological structure exists within individuals' aesthetic psychological structures, and this is a structure or organisation of general mental activity conducive to "aestheticisation", i.e., one's "aesthetic tendencies" that essentially govern, regulate and interpret one's

aesthetic psychology. The five aesthetic value tendencies of university students therefore correspond to the five factors of personality, wherein their aesthetic personalities constitute stable value tendencies and habitual ways of behaving regarding aesthetic objects formed by individuals in their aesthetic practices. According to Zheng et al. [34], individuals have different value tendencies, such as "pursuing harmony", "focusing on appreciation", "promoting fashion", "obsessing over tragedy", or "behaving differently". Zhao Yongping's study also shows that university students have five important aesthetic tendencies, namely, a pursuit of harmony, an emphasis on appreciation, an esteem for fashion, an obsession with tragedy and the display of difference [35]. Second, the aesthetic tendency of showing difference is consistent with the characteristics of university students, while they value the factor of being obsessed with tragic emotions the least. Similarly, Cui Meiyu and Zhu Caixia find that the factor of "focusing on appreciation" is the most valued by university students, who focus on the appreciation of beauty in terms of appearance, vulgarity and figurativeness; the second most important aesthetic value tendency of university students is "showing difference", which reflects how contemporary university students emphasise individual difference and pursue individuality in their aesthetic values [36]. Hence, the above analysis shows that aesthetic values have individual differences and that the five major aesthetic tendencies of college students correspond with the Big Five factors of personality.

Furthermore, individuals form values in their lives and practices to govern their judgements and choices and determine their attitudes and behaviours [37]. Aesthetic value is an important part of life value and involves people's awareness of the aesthetic value of objective things and phenomena and is reflected in their judgements, evaluations and behavioural tendencies concerning things from an aesthetic perspective. Generally, it regards beauty and ugliness, which relate to what people like or dislike—the basic perspective from which people distinguish beauty from ugliness [38]. Aesthetic value is therefore formed based on aesthetic needs and amid the interaction of various psychological functions, such as aesthetic cognition and aesthetic experience [39].

According to the theory of planned behaviour, behavioural intention is an important aspect in predicting behaviour, and values directly influence individuals' judgements and choices among various values, things and behaviours [40]. In a study of the causes and countermeasures affecting the development of extracurricular physical activities in colleges and universities, Zhao Mei points out that students' incorrect perceptions of sports are the root cause of university students' participation in extracurricular physical activities [41]. Values are the perceptions that arise during people's daily lives and activities that guide their judgements and choices and in turn influence their attitudes and actions [37]. Based on attitude change theory, Tang Hongxing has thus suggested that changes in attitudes towards exercise can significantly improve individuals' exercise behaviours [42]. Zhao Lingli conducted an empirical study on the role of aesthetic education in the formation and transformation of young people's life values, concluding that aesthetic education can have a positive impact on young students' life values and that long-term education may have a comprehensive impact on people's values and thus drive them to change their behaviours. Hence, aesthetic education enables young people to maintain a positive psychological state and to be receptive to education [43]. Xu Zhen has also shown that through the study of beauty, people's cognitive and aesthetic experiences can be enhanced, which in turn improves their overall well-being, thereby contributing to the development of a sound personality in individuals.

By using functional magnetic resonance imaging (fMRI) to examine the aesthetic judgements of a group of architects and a group of nonarchitects regarding visually presented architectural stimuli and control stimuli (faces), Kirk et al. confirmed that expertise modulates neural activity in the brain regions associated with cognitive processing and reward processing

[44]. Another experiment showed that there was little difference in the aesthetic judgements of professionals for artworks labelled with different monetary values but the same content and observed no difference in the activation of their ventral medial prefrontal lobes, which was tested and found to be calibrated by expertise [45]. Then, The richness of the individual's expertise can significantly influence an individual's judgements and choice of aesthetic objects, with experts paying more attention to and analysing the form and content thereof and nonexperts being more influenced by external factors. Professional training and experience thus enhance the sensitivity to graphic organisation and the extraction of the aesthetic features of artworks.

Thus, this study's third hypothesis is as follows:

H3: The aesthetic cognitions of university students significantly and positively influence their aesthetic values.

Furthermore, Strobl and Grail argue that attitude is a mediating variable between knowledge and behaviour, individual behaviour is influenced by many factors, of which attitude is one that has an internal influence on individual behaviour, resulting in lasting, consistent change [46]. Aesthetic activity is thus a cognitive experiential activity in human society. Aesthetic cognition is also an important prerequisite for the implementation of aesthetic behaviour. According to the cognitive-affect-behavioural model [47], decision-making begins with cognition, which is followed by emotion and finally behaviour. In other words, cognition determines emotion, which in turn leads to behaviour; more importantly, emotion regulates the relationship between cognition and behaviour.

Aesthetic cognition represents the influence of aesthetic values through aesthetic emotions (positive emotions). Aesthetics motivate individuals to develop positive values in life, thus contributing to improvement in their aesthetic behaviour. Aesthetic expertise has a "corrective" effect on aesthetic values, as a wealth of the former enables individuals to make professional judgements. Meanwhile, their positive emotions will lead these individuals to engage in aesthetic creativity and thus exhibit better creative behaviour. That is, aesthetic emotion is a positive emotion, and all positive emotions promote creativity.

Thus, this study's fourth hypothesis is as follows:

H4: The aesthetic emotions and aesthetic value tendencies of university students have a chain mediating effect on the relationship between their aesthetic cognitions and aesthetic behaviours.

## 2 Methods

Written informed consent was given from all the participants prior to this study in accordance with the Declaration of Helsinki. Written informed consent was obtained from all the participants prior to this study.

### 2.1 Participants

From 1 July 2022 to 8 July 2022, the researcher used Questionnaire Star software to distribute the questionnaires on the online platform, and 300 college students from five colleges and universities in Shaanxi Province, China, were selected as the official participants of the pretest for this study. The study distributed the official questionnaire to the participants three weeks after the pre-test questionnaire testing was completed and revised, the participants in the study were between the ages of 18 and 28 years old, and each participant received a cash prize for completing the questionnaire through the online platform. Written informed consent was obtained from all the participants prior to this study.

A total of 1132 participants were received, with a total of 1060 valid questionnaires and a recovery rate of 93.6%. The source of formal participants are Shaanxi Normal University, Northwestern Polytechnical University, Xi'an University of Posts & Telecommunications, Northwest University, Northwest University of Political Science and Law, Xi'an International Studies University, Xidian University,Yulin Vocational and Technical College,Weinan Normal University, Yulin University,Xi'an University of Finance and Economics.The participants consisted of 589 (55.6%) males, 471 (44.4%) females, 422 (39.8%) art students, and 638 (60.2%) science students.

## 2.2 Material

**Aesthetic cognition.**   The aesthetic general knowledge scale developed by Wu Guanxian [48] was used in this study. This questionnaire consists of 11 questions across 2 dimensions, including 6 questions on aesthetic knowledge and 5 questions on appreciation and expression. Aesthetic knowledge questions, such as "I understand the meaning of artistic styles (including visual arts, music, performance)", and appreciation and performance questions, such as "I play at least one instrument ", were scored on a five-point positive scale (1 = very unlikely, 5 = very likely), with higher scores indicating higher levels of aesthetic cognition.

**Aesthetic emotion.**   The aesthetic affect scale developed by Diessner et al. [49] was used in this study. This questionnaire consists of 10 questions and two dimensions; there are three questions on natural beauty and seven questions on human beauty (artistic and moral beauty). The natural beauty questions include "When I perceive the beauty of nature, I feel a sense of union with the universe or a love for the whole world". Human beauty questions include "When I perceive the beauty of a work of art, I feel a sense of awe, wonder, excitement, admiration, and upward mobility". The questionnaire was scored on a five-point positive scale (1 = very unlikely, 5 = very likely), with higher scores indicating a higher level of aesthetic experience.

**Aesthetic tendencies.**   The aesthetic tendency scale designed by Zheng Chung et al. [34] was adopted in this study. The final design of the aesthetic tendency scale is based on two dimensions, namely, the pursuit of harmony and appreciation, with three questions for each of them and six questions in total. Appreciation-oriented questions include "I think that beauty is always far-reaching and evocative". The questionnaire was scored on a five-point positive scale (1 = very unconforming, 5 = very conforming), with higher scores indicating a more positive cognition of aesthetic values.

**Aesthetic behaviour.**   In this study, aesthetic behaviour is divided into three dimensions: the behavioural habit of participating in art, the beautification of the living environment, and the reasonable matching of clothing. The aesthetic behaviour scale designed by Wu Guanxian[48] and Yi [8] was used, Wu Guanxian was referenced for the dimension of participation in art, and Yi was referenced for the dimensions of beautification of the environment and matching of clothing. The questionnaire contains 11 questions and three dimensions, namely, four questions on the behavioural habits of participating in art, four on the beautification of the living environment, and three on the rational matching of clothing, such as "I often visit art exhibitions (painting, heritage, music, dance, theatre)", "I like to beautify my living or working space with flowers, plants, artwork or other decorative objects", and "I know which colours and styles of clothing suit me". The questionnaire was scored on a five-point positive scale (1 = very unsuitable, 5 = very suitable), with higher scores indicating more positive daily aesthetic behaviour.

## 2.3 Confidence and validity analysis

In this study, SPSS 23 and AMOS 24 were used to verify the reliability and validity of the collected scale data, and structural equation models were used to construct measurement models [50,51].

**Table 1. Fit test data for the scales.**

| Model | $\chi2/df$ | RMSEA | RMR | AGFI | CFI | NFI | IFI | TLI |
|---|---|---|---|---|---|---|---|---|
| Standard | < 3 | < 0.08 | < 0.05 | >0.9 | >0.9 | >0.9 | >0.9 | >0.9 |
| aesthetic cognition | 2.757 | 0.041 | 0.016 | 0.969 | 0.993 | 0.989 | 0.993 | 0.988 |
| aesthetic emotions | 5.997 | 0.069 | 0.019 | 0.940 | 0.984 | 0.981 | 0.984 | 0.971 |
| aesthetic tendencies | 8.030 | 0.081 | 0.029 | 0.948 | 0.979 | 0.976 | 0.979 | 0.954 |
| aesthetic behaviour | 3.707 | 0.051 | 0.018 | 0.960 | 0.985 | 0.979 | 0.985 | 0.977 |

The variance of error for the aesthetic cognition was between 0.153 and 0.641, and the results were positive and significant. All standardized regression coefficients ranged from 0.61 to 0.9, with no coefficients above or overly close to 1.The stimated standard errors (SEs) of the variance of measurement errors ranged from 0.013 to 0.035, with no considerable SE observed. The variance of error for the aesthetic emotion was between 0.167 and 0.520, and the results were positive and significant. All standardized regression coefficients ranged from 0.632 to 0.890, with no coefficients above or overly close to 1.The stimated standard errors (SEs) of the variance of measurement errors ranged from 0.014 to 0.025, with no considerable SE observed.The variance of error for the aesthetic tendencies was between 0.195 and 0.505, and the results were positive and significant. All standardized regression coefficients ranged from 0.641 to 0.847, with no coefficients above or overly close to 1.The stimated standard errors (SEs) of the variance of measurement errors ranged from 0.018 to 0.029, with no considerable SE observed. The variance of error for the aesthetic behaviour was between 0.205 and 0.418, and the results were positive and significant. All standardized regression coefficients ranged from 0.679 to 0.845, with no coefficients above or overly close to 1.The stimated standard errors (SEs) of the variance of measurement errors ranged from 0.014 to 0.024, with no considerable SE observed.

The results of the fitting tests for the four scales indicate that all values were within an acceptable range (presented in Table 1). Confirmatory factor analysis results indicated that the four scales had sufficient validity [52].

Tests for convergent validity indicated that the standardized factor loadings of the aesthetic cognition scale, aesthetic emotion scale, aesthetic tendencies scale, and aesthetic behaviour scale were in the range of 0.61–0.9, 0.632–0.890, 0.641–0.847, and 0.679–0.845 respectively; these values were all greater than the acceptable criterion of 0.5 and were all significant [53,54]. The combined reliability values for each dimension of each scale range from 0.781 to 0.915, all reaching the standard of being higher than 0.6 [55]. Average variant extraction (AVE) values ranged from 0.493 to 0.76, meeting the standard of being higher than 0.5 [55]. Therefore, the professional identity scale has sufficient convergent validity. The discriminant validity test indicated that the correlation coefficient of each dimension of the four scales was between 0.273 and 0.718, and a significant correlation was observed. The square root of AVE of each dimension of the scale was between 0.702 and 0.872, and the correlation coefficient value of each dimension was less than the square root of the AVE. And Cronbach's coefficients for the four scales ranged between 0.815 and 0.933. This indicates a certain correlation and a certain degree of discrimination between the latent variables. This also indicates that the four scales had sufficient discriminant validity [55].

## 2.4 Common method deviation test

In this study, after the questionnaires were collected, collated and tested using Harman's one-way method, all the variables measured were included in one exploratory item for analysis,

**Table 2. Summary table of the correlation analysis between aesthetic cognition, aesthetic emotion, aesthetic tendency and aesthetic behaviour as a whole.**

| Variables | M | SD | 1 | 2 | 3 | 4 |
|---|---|---|---|---|---|---|
| Aesthetic cognition | 3.28 | 0.69 | 1 | | | |
| Aesthetic emotion | 3.62 | 0.70 | 0.61** | 1 | | |
| Aesthetic tendencies | 3.52 | 0.66 | 0.46** | 0.63** | 1 | |
| Aesthetic behaviour | 3.46 | 0.64 | 0.64** | 0.66** | 0.57** | 1 |

and a total of seven principal components were extracted before factor rotation. The first factor explained 40.17% of the variance, which was less than 50%, thus indicating that these data had no problem with common method bias and could thus be analysed in depth [56].

# 3 Results

## 3.1 Descriptive analysis results

Table 2 presents the results of the descriptive statistics and correlation analysis for each variable. These results indicate that the current aesthetic cognitions, aesthetic affect, aesthetic tendencies and aesthetic behaviours of the university students were at a moderate to high level. There is a significant positive correlation between the aesthetic cognition, aesthetic emotion, aesthetic value tendency and aesthetic behaviour variables, and the data from this study can be used in further chain-mediated effect analysis.

## 3.2 Main effect

As shown in Table 3, the regression analyses revealed that after controlling for the effects of sex and and subject, aesthetic cognition directly and positively predicts aesthetic affect ($\beta$ = 0.60, p<0.001); aesthetic cognition directly and positively predicts aesthetic value tendency ($\beta$ = 0.12, p<0.001); and aesthetic affect directly and positively predicts aesthetic tendency ($\beta$ = 0.56, p<0.001). Moreover, aesthetic cognition, aesthetic emotion and aesthetic value tendency simultaneously predict aesthetic behaviour, aesthetic cognition, aesthetic emotion and aesthetic value tendency, as all have a direct positive predictive effect on aesthetic behaviour ($\beta$ = 0.36, p<0.001; $\beta$ = 0.29, p<0.001; $\beta$ = 0.22, p<0.001).

**Table 3. Regression analysis between variables.**

| Regression equation | | | Overall fit index | | | Significance of regression coefficients | |
|---|---|---|---|---|---|---|---|
| Resulting variables | Predictor variables | R | R² | F | $\beta$ | t | |
| Aesthetic emotion | Sex | 0.62 | 0.38 | 215.52 | 0.07 | 2.75** | |
| | Subject | | | | -0.04 | -1.59 | |
| | Aesthetic cognition | | | | 0.60 | 24.78*** | |
| Aesthetic tendencies | Sex | 0.64 | 0.41 | 179.48 | -0.05 | -2.11* | |
| | Subject | | | | -0.00 | -0.09 | |
| | Aesthetic cognition | | | | 0.12 | 3.90*** | |
| | Aesthetic emotion | | | | 0.56 | 18.56*** | |
| Aesthetic behaviour | Sex | 0.75 | 0.56 | 263.32 | 0.04 | 1.70 | |
| | Subjects | | | | -0.04 | -1.80 | |
| | Aesthetic cognition | | | | 0.36 | 13.65*** | |
| | Aesthetic emotion | | | | 0.29 | 9.67*** | |
| | Aesthetic tendencies | | | | 0.22 | 8.37*** | |

### 3.3 Structural model

The various fitted indicators in the chain-mediated analysis measurement model of aesthetic emotion and aesthetic value tendency showed that the $\chi2/df$ indicator did not meet the standard. Through continuous correction, the fitted model was finally obtained with the following values indicating excellent fit: $\chi2/df$ of 6.18, RMSEA of 0.07, AGFI of 0.94, NFI of 0.98, TLI of 0.96 and CFI of 0.98.The fit metrics were excellent and all fitness metrics met the standard [51,57,58].

### 3.4 Mediating effects

This study used the bootstrap method, repeating the sampling 2000 times to calculate confidence intervals at the 95% level of bias correction, to test the significance of the mediation effect. Bootstrapping was used to find the trust interval for the indirect effect, where if this trust interval did not contain 0, then the mediated effect was said to be present [59]. As shown in Table 4, first, the combined effect of university students' aesthetic cognitions on their aesthetic behaviours was analysed to test whether there is a direct correlation between these two variables or whether their relationship is mediated by other variables. The 95% confidence interval for the total effect is [0.65, 0.79], which does not contain 0, indicating that the total effect of aesthetic cognition on aesthetic behaviour is significant. The 95% confidence interval for the total indirect effect is [0.27, 0.50], which does not contain 0, indicating that the indirect effect of aesthetic cognition on aesthetic behaviour is significant. The 95% confidence interval for the direct effect is [0.21, 0.48], which does not contain 0, indicating that the indirect effect of aesthetic cognition on aesthetic behaviour is significant. Finally, the 95% confidence interval for the direct effect is [0.21, 0.48], which does not contain 0, indicating that the indirect effect of aesthetic cognition on aesthetic behaviour is significant. In summary, these results show that the total, indirect and direct effects of aesthetic cognition on aesthetic behaviour are significant and that the effect of aesthetic cognition on aesthetic behaviour is partially mediated.

Further analysis showed that the influence of aesthetic cognition on aesthetic behaviour is influenced by three indirect effects, one of which reached significance. First, path 1 (0.24), consisting of aesthetic cognition → aesthetic emotion → aesthetic value tendency → aesthetic

**Table 4. Bootstrap method estimates 95% confidence interval.**

| | Path | Path Coefficient | BC(Bias-Corrected) | |
|---|---|---|---|---|
| | | | Lower | Upper |
| Total effect | Aesthetic cognition →Aesthetic behaviour | 0.72*** | 0.65 | 0.79 |
| Direct effect | Aesthetic cognition →Aesthetic behaviour | 0.35*** | 0.21 | 0.48 |
| Total indirect effect | Aesthetic cognition →Aesthetic behaviour | 0.38*** | 0.27 | 0.50 |
| Indirect effect1 | Aesthetic cognition →Aesthetic emotion →Aesthetic tendencies→Aesthetic behaviour | 0.24*** | 0.11 | 0.44 |
| Indirect effect2 | Aesthetic cognition →Aesthetic emotion →Aesthetic behaviour | 0.14 | -0.08 | 0.30 |
| Indirect effect3 | Aesthetic cognition →Aesthetic tendencies→Aesthetic behaviour | -0.01 | -0.09 | 0.04 |

***$p < 0.001$.

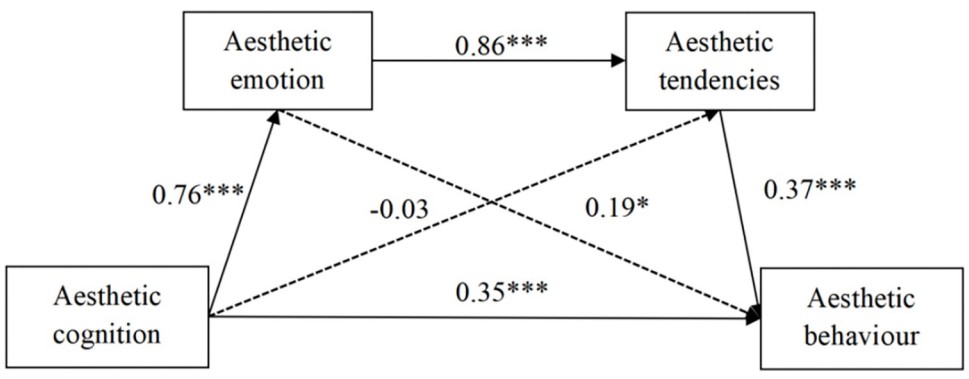

**Fig 1. Structural model.**

behaviour, with a confidence interval [0.11, 0.44], did not contain 0, indicating that aesthetic emotion and aesthetic value tendency have a significant chain mediating effect on the relationship between aesthetic cognition and aesthetic behaviour. Hence, H4holds(presented in Fig 1). Second, path 2 (0.14), consisting of aesthetic cognition → aesthetic emotion → aesthetic behaviour, with a confidence interval [-0.08, 0.30], contains 0, indicating that the mediating effect of aesthetic emotion in the relationship between aesthetic cognition and aesthetic behaviour is not significant. Third, path 3 (-0.01), consisting of aesthetic cognition → aesthetic tendency → aesthetic behaviour confidence interval [-0.09, 0.04], contains 0, indicating that the mediating role of aesthetic value tendency in the relationship between aesthetic cognition and aesthetic behaviour is not significant.

## 4 Discussion

This study found that university students' aesthetic cognitions have a significant positive impact on their aesthetic emotions, affirming H1. These results are consistent with cognitive-emotional theory [10,11] and with the findings of scholars [17]. The importance of aesthetic understanding in aesthetic experience is undisputed, and the successful completion of each stage of the aesthetic process triggers an emotional experience of self-reward and continuously increases the overall level of a subject's aesthetic emotions; these results are consistent with previous findings [20–24].

This study also found that university students' aesthetic emotions have a significant positive impact on their aesthetic behaviours, affirming H2. These results echo the extended-constructivity theory of positive emotions [28]. In line with attribution theory, people with positive emotions are more likely to attribute their failures to external circumstances [31]. Such individuals maintain a higher level of creative efficacy and thus have more confidence in their subsequent creative work. Positive emotions lead to more creativity and flexibility in thinking and acting, playing a more significant role in creativity [29,30].

Moreover, this study found that university students' aesthetic perceptions have a significant positive impact on their aesthetic value tendencies, affirming H3.

This result thus indirectly confirms the findings of Kirk et al. [44], who found that professional knowledge significantly influences individual aesthetic judgements. The above results suggest that aesthetic perceptions significantly influence aesthetic value tendencies (focusing on appreciation versus the pursuit of harmony), and these findings echo the calibrating effect of expertise on aesthetic judgements [45]. The categorisation of artworks reflects one's greater knowledge of art history, whereby one does not rely solely on the superficial features of

artworks to make judgements. When appreciating paintings, they are thus sensitive to colours, lines and shapes, which indicates that they apply professional aesthetic judgement.

Finally, this study found that university students' aesthetic emotions and aesthetic tendencies have a chain mediating effect on the relationship between their aesthetic perceptions and aesthetic behaviours (H4). It is thus evident that the aesthetic cognitions of university students must indirectly influence their aesthetic behaviours through their aesthetic emotions and aesthetic value tendencies. In line with the literature [46], these findings are also consistent with the cognitive-affective-behavioural model [47]; values are the perceptions that emerge during daily life and activities that guide people's judgements and choices and in turn influence their attitudes and actions. These findings are consistent with research [42,43] suggesting that their understanding and judgements of beauty and their preferences and choices of aesthetic interest influence people's aesthetic behaviours.

## 5 Conclusion

College students' aesthetic cognitions impact their aesthetic behaviour intentions through their aesthetic emotions, while positive and pleasant emotions promote individuals' preferences for harmonious beauty. College students thus pay attention to appreciation and comprehension when appreciating aesthetics; when appreciating paintings, they are more sensitive to colour, line and composition, which shows that their aesthetic cognitions influence their aesthetic preferences or tendencies through their aesthetic emotions. "The pursuit of harmonious interest" is a positive aesthetic tendency, and university students love harmonious and elegant scenes and believe that the enjoyment of beauty should be relaxing and pleasant, which is a positive and healthy aesthetic interest. Concerning the role played by aesthetic factors in personality, they not only are a component of sound personality but also can act as a mediating force promoting perfection and the improvement of the intelligent and moral factors in one's personality structure. In summary, this study takes the aesthetic behaviours and habits of university students as the object (including their behaviours and habits in participating in art, the beautification of their home environment, and their reasonable matching of clothing) and constructs a psychological behavioural mechanism model of university students' aesthetic behaviours in terms of their levels of aesthetic cognition, aesthetic emotion and aesthetic tendency. The study therefore reveals the role of aesthetic emotion and aesthetic tendency in a chain mediation of the relationship between aesthetic cognition and aesthetic behaviour. This study reveals the chain-mediating effects of university students' aesthetic affect and aesthetic disposition between their aesthetic perceptions and aesthetic behaviour.

## 6 Research limitations

### 6.1 Research sample

In terms of sampling, due to limitations of the researcher's time and ability, the sample of this study consisted only of students drawn from several universities in Shaanxi Province, China, and it is not possible to account for more Chinese college students in other provinces, thus limiting the interpretation of the results of the study and the inferences that can be drawn based on those results.

### 6.2 Research variables

Many factors influence Chinese college students' aesthetic behaviour, and this study is limited to three influences: aesthetic cognition, aesthetic emotion and aesthetic tendency. Other possible influencing factors, such as Chinese college students' personality traits, parents' income

level, parents' attitudes towards art, the economic and cultural level of their cities, and the aesthetic environment, are not included in the variables explored in this study.

## 6.3 Research method

This study uses a questionnaire survey to collect information about the personal background variables, aesthetic cognition, aesthetic emotion aesthetic tendency and aesthetic behaviour of Chinese college students in Shaanxi Province; however, concerns regarding whether these subjects can truly express and reflect the real situation may bias the correctness of the results of the study.

## 7 Research suggestions

First, the cultivation of cognitive ability is an effective way to change an individual's original way of perceiving and expressing art, and discerning the laws of art via rational thinking is an effective way to promote improved aesthetic ability. In the implementation of aesthetic education, the acquisition of aesthetic knowledge and the improvement in cognitive ability are important ways to promote the formation of a noble personality. To appreciate the beauty of things, we must have a mind capable of perceiving beauty, and for this perception to be highly rational and active, it must be cultivated through theoretical learning and knowledge. Therefore, there is a need to strengthen the systematic understanding and learning of the history of aesthetics and aesthetic knowledge and skills, this will not only allow the aesthetic subject to appreciate art effectively but also help him or her resonate with the life and personality of creators and thus engage in a higher aesthetic experience.

Second, Works of art are born on the basis of objective reality and the experiences of their creators and are conscious and regular creations. Therefore, in the process of implementing aesthetic education, the acquisition of aesthetic knowledge and the cognitive ability to improve aesthetic and emotional abilities and promote individuals in forming a noble personality is important. According to the Marxist view, beauty is the unity of subjectivity and objectivity. To appreciate the beauty of things, we must have a mind capable of perceiving beauty, and for this perception to be highly rational and active, it must be cultivated through theoretical learning and knowledge. Aesthetic education is the cultivation of aesthetic sensibility based on human sense organs. The stronger the subject's ability to recognise objective reality, the higher the level of understanding of the social value of art and the aesthetic elements contained in art. Taking visual aesthetic education as an example, Aesthetic quality is first based on the cultivation of visual aesthetic perception, which depends on students' visual perceptions, theoretical cognitions of visual aesthetics and practical experiences of visual expression. Cognitive and practical experiences act on the body's visual cognitive nervous system, forming excitatory neurons in the relevant areas of the brain to achieve aesthetic cognition. In the teaching process, teachers need to have a precise grasp of representative aesthetic materials and present them as a means to promote the formation of students' aesthetic cognitive abilities and enhance their levels of aesthetic experience. Second, the aesthetic levels of students should be enhanced based on emotion and reason so that they can use the norms of beauty to observe and create and thus establish correct aesthetic values.

Third, a beautiful and harmonious campus culture should be created. School infrastructure and campus environment are an important basis for aesthetic education and aesthetic activities. The green environment of the campus, campus buildings, campus humanistic environment and classroom indoor environmental facilities are all aesthetic objects that students can visually perceive. These natural scenes and the humanistic environment together form the aesthetic environment of the campus. The construction of the campus environment should follow

the principle of aesthetics, reflecting the elegant beauty of art and highlighting the beauty, sublimity and magnificence of art. The sculptures and paintings in the cultural environment of the campus are important visual elements that enrich the spiritual lives of teachers and students, and the design of the campus cultural environment can reflect the spiritual pursuit of teachers and students. Listening to classical music, viewing classical paintings and sculptures, and appreciating classical literature can help students develop their aesthetic interests and ideals. With an elegant aesthetic ideal, they will learn and follow the example of the aesthetic ideal in their hearts, and they will naturally turn away from vulgarity and develop aesthetic behaviour.

Fourth, universities should provide various forms of rich aesthetic activities and encourage university students to actively participate in aesthetic activities. Rich aesthetic practices can help students understand the importance and significance of aesthetics. The aesthetic practice of university students involves their aesthetic activities, in which the subjects reach the "world of meaning" and "world of value" through sensory, cognitive and emotional life experiences based on their own personal experiences and eventually form an attitude towards beauty. Subjects' attitudes towards beauty are formed through their sensory, cognitive and emotional life experiences. Students should actively participate in the various activities organised by art societies in universities, such as book clubs, lecture societies and literary societies, to discover and experience the beauty of art through reading, film reviews and concerts. In the process of cultivating aesthetic literacy, there is an essential difference between active participation and passive appreciation. Beauty is creative, and creation involves participation; participation is the essence of art, and operation is the way art exists; only in the process of practical operation can the spirit of beauty and art be truly appreciated, enabling students to enter the depths of art and cultivate their aesthetic literacy to move towards an advanced form.

## Supporting information

**S1 Data.**
(SAV)

## Acknowledgments

We would like to express our gratitude to all who participated in the study.

## Author Contributions

**Conceptualization:** Qiao Qiao, Yongzhi Jiang.

**Data curation:** Qiao Qiao, Yongzhi Jiang.

**Formal analysis:** Qiao Qiao, Yongzhi Jiang.

**Funding acquisition:** Qiao Qiao.

**Investigation:** Qiao Qiao.

**Methodology:** Qiao Qiao.

**Project administration:** Qiao Qiao.

**Resources:** Qiao Qiao.

**Software:** Qiao Qiao.

**Supervision:** Qiao Qiao.

**Validation:** Qiao Qiao.

**Visualization:** Qiao Qiao.

**Writing – original draft:** Qiao Qiao.

**Writing – review & editing:** Qiao Qiao.

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
