## [Decision Letter · Decision Letter 0]

25 Aug 2023

PONE-D-23-22731The Influence of College Students' Aesthetic Cognitions on Aesthetic Behaviours: The Chain Mediation EffectPLOS ONE

Dear Dr. Jiang,

Thank you for submitting your manuscript to PLOS ONE. After careful consideration, we feel that it has merit but does not fully meet PLOS ONE’s publication criteria as it currently stands. Therefore, we invite you to submit a revised version of the manuscript that addresses the points raised during the review process.

We look forward to receiving your revised manuscript.

Kind regards,

Maja Vukadinovic

Academic Editor

PLOS ONE

Additional Editor Comments:

I have concerns regarding Ethics Statement. T is written :'' Informed consent.

The participants had the study purpose explained to them first, and then they were

asked to provide written informed consent. Participation was voluntary, and all data

were handled confidentially.

Please include these information in Participants section.

Reviewers' comments:

Reviewer's Responses to Questions

**Comments to the Author**

1. Is the manuscript technically sound, and do the data support the conclusions?

Reviewer #1: Yes

Reviewer #2: Yes

Reviewer #3: Partly

2. Has the statistical analysis been performed appropriately and rigorously? 

Reviewer #1: Yes

Reviewer #2: Yes

Reviewer #3: Yes

3. Have the authors made all data underlying the findings in their manuscript fully available?

Reviewer #1: Yes

Reviewer #2: Yes

Reviewer #3: Yes

4. Is the manuscript presented in an intelligible fashion and written in standard English?

Reviewer #1: No

Reviewer #2: Yes

Reviewer #3: No

5. Review Comments to the Author

Reviewer #1: I found the introduction, while informative, too long. Perhaps, it can be reduced to a more reasonable size without losing the justification for each hypothesis.

Method-Participants section: “A total of 1132 samples were received, … Among the 1060 official samples, ..” I believe you mean participants, not samples!

Furthermore, the writing needs to be a bit tighter. For example, the same paragraph should read: The sample consisted of 589 (55.6%) males, 471 (44.4%) females (44.4%), 422 (39.8%) art students, and 638 (60.2%) science students.

Materials (Instruments): Please provide the validity and reliability data of the instruments used in this work (aesthetic general knowledge scale, aesthetic affect scale, aesthetic tendency scale, aesthetic behaviour scale.

What is the purpose of this study? Is one of the aims of this study to provide validity and reliability measures of these instruments? If so, is this confirmation of what already exists and why?

Are you validating these instruments or exploring the relationships between what these instruments measure?

I found the result section, especially the part referring to factor analysis, hard to follow. Several statistical parameters are listed without explanation. These parameters should be listed in an appropriate table with the factor loadings and reference to rotation, if any. This section should be written more clearly with an emphasis on providing a sense of purpose, avoiding unnecessary details (i.e., statistical parameters) within the text, and relating the information gained from these analyses to what comes next and the big picture. The details of the factor analysis should be presented in a table, while the text should be reserved for reporting the essential details.

I assume the reference to “gender” to mean “sex.”

The statement “through statistical analysis” is unnecessary!

A section discussing the limitations of this study and their implication is missing.

Reviewer #2: This manuscript addresses a very interesting topic, not commonly addressed in higher education research and from a quantitative perspective. In this paper, the philosophical concept of aesthetic is discussed in the light of the relationship between psychological and behavioral students´characteristics. The study presents a complete literature review, however, sometimes, it lacks from the critical voices of the authors.

The methods section should describe the sample in a more detailed way (characteristics, access, selection criteria, informed consent, etc.). Results are presented in a clear and consistent manner. Summarizing, the manuscript presents a sound piece of scientific research with data supporting findings and sound conclusions. I recommend the paper for publication with minor corrections (Literature review, sample).

Reviewer #3: Abstract:

It would be good to include some broader context into this abstract. Why is important to study students’ aesthetic behaviors? What is the purpose of the four scales? What is a daily aesthetic practice, and why would the reader want to know this? I can see from the intro that this is well developed in Chinese education, but it would be helpful for an international audience to have a bit more background.

Introduction:

• The introduction is quite long (2900 words), and reads a bit like a thesis. I think it can be streamlined and focused to bring the reader into the paper without having to explore each hypothesis in such detail.

• Beware of paragraphs that are too long, especially the first one, which can be difficult for the reader to follow.

• Early in the intro, it would be helpful to define aesthetic education.

Methods:

• It would help to know more about the participants. How many schools? What kinds of schools and what geographic area? What levels? How were they selected?

• There appears to be some results mixed into the methods section.

Discussion and Conclusion

• I liked these sections better. They were a bit more focused and brought things into a broader context. I liked the connection to specific recommendations for universities.

6. PLOS authors have the option to publish the peer review history of their article (what does this mean?). If published, this will include your full peer review and any attached files.

Reviewer #1: **Yes: **Dr. Ali M AL-Asadi

Reviewer #2: No

Reviewer #3: No

---

## [Author Response · Author response to Decision Letter 0]

15 Sep 2023

Dear reviewer,

 Thank you very much for your comments and professional advice. These opinions help to improve academic rigor of our article.Based on your suggestion and request, we have made corrected modifications on the revised manuscript, Meanwhile, the manuscript had be reviewed and edited by language servicices of AJE SEVIER. We hope that our work can be improved again. Furthermore, we would like to show the details as follows:

Reviewer #1: I found the introduction, while informative, too long. Perhaps, it can be reduced to a more reasonable size without losing the justification for each hypothesis.

Method-Participants section: “A total of 1132 samples were received, … Among the 1060 official samples, ..” I believe you mean participants, not samples!

Furthermore, the writing needs to be a bit tighter. For example, the same paragraph should read: The sample consisted of 589 (55.6%) males, 471 (44.4%) females (44.4%), 422 (39.8%) art students, and 638 (60.2%) science students.

Materials (Instruments): Please provide the validity and reliability data of the instruments used in this work (aesthetic general knowledge scale, aesthetic affect scale, aesthetic tendency scale, aesthetic behaviour scale.

What is the purpose of this study? Is one of the aims of this study to provide validity and reliability measures of these instruments? If so, is this confirmation of what already exists and why?

Are you validating these instruments or exploring the relationships between what these instruments measure?

I found the result section, especially the part referring to factor analysis, hard to follow. Several statistical parameters are listed without explanation. These parameters should be listed in an appropriate table with the factor loadings and reference to rotation, if any. This section should be written more clearly with an emphasis on providing a sense of purpose, avoiding unnecessary details (i.e., statistical parameters) within the text, and relating the information gained from these analyses to what comes next and the big picture. The details of the factor analysis should be presented in a table, while the text should be reserved for reporting the essential details.

I assume the reference to “gender” to mean “sex.”

The statement “through statistical analysis” is unnecessary!

A section discussing the limitations of this study and their implication is missing.

The author’s answer: We have revised the sentence in the article. The details as follows:

Firstly,The introduction section has cut the number of words (2372 words) without losing the justification for each hypothesis.

“In recent years, the development of aesthetic education has received a great deal of attention in Chinese education. Aesthetic education aims to improve the aesthetic qualities of Chinese university students and has positive significance in cultivating noble aesthetic pursuits and noble personalities among university students. Aesthetic behaviour is the outwards manifestation of aesthetic literacy, aesthetic common sense and aesthetic concepts, which constitute the aesthetic qualities of individuals; these can be concretely expressed only through aesthetic behaviour [1]. Aesthetic education evokes and forms characteristics and attributes with human value in students [2]. However, some scholars claim that aesthetic education is not effective [3-6]. Shi O and Hou Jingmin [7] suggest that there is a substantial difference between passive viewing and active participation in the creation of beauty for improving aesthetic literacy, that students' aesthetic literacy is not only reflected in their artistic activities, that aesthetics should be infused into life, that the act of aesthetics is a form of aesthetic creativity, and that this ability to create beauty is expressed in the ability to create and express using diverse materials and multiple methods [8]. This study posits that the aesthetic practice behaviours of university students can be divided into three aspects: the behavioural habit of participating in art, the beautification of the living environment, and the rational matching of clothing，moreover, the three aspects represent the fit and communication between the aesthetic subject and object, which comprehensively reflects the quality of human existence and is the most typical mass aesthetic activity. This involves a kind of creation in which the relationship between the subject and object is reciprocal and constitutes the action of the subject in consciously interacting with the aesthetic object to achieve pleasure and creation in life [9]. There is very little research on the everyday aesthetic behaviour of university students. This study uses structural equation modelling to conduct an empirical study. The study of the psycho-behavioural mechanisms of the aesthetic behaviours of university students is important for understanding the characteristics of aesthetic psychology and ensuring high-quality development of aesthetic education”. 

Secondly, The “samples” has been changed to the “participants”.

Method-Participants section: Written informed consent was given from all the participants prior to this study in accordance with the Declaration of Helsinki. Written informed consent was obtained from all the participants prior to this study.From 1 July 2022 to 8 July 2022, the researcher used Questionnaire Star software to distribute the questionnaires on the online platform, and 300 college students from five colleges and universities in Shaanxi Province, China, were selected as the official participants of the pretest for this study. The study distributed the official questionnaire to the participants three weeks after the pre-test questionnaire testing was completed and revised, the participants in the study were between the ages of 18 and 28 years old, and each participant received a cash prize for completing the questionnaire through the online platform. Written informed consent was obtained from all the participants prior to this study. 

A total of 1132 participants were received, with a total of 1060 valid questionnaires and a recovery rate of 93.6%. The source of formal participants are Shaanxi Normal University, Northwestern Polytechnical University, Xi’an University of Posts & Telecommunications, Northwest University, Northwest University of Political Science and Law, Xi`an International Studies University, Xidian University,Yulin Vocational and Technical College,Weinan Normal University,Yulin University,Xi’an University of Finance and Economics.The participants consisted of 589 (55.6%) males, 471 (44.4%) females, 422 (39.8%) art students, and 638 (60.2%) science students.

Materials (Instruments):This part has been modified to increase the reliability and validity.

“In this study, SPSS 23 and AMOS 24 were used to verify the reliability and validity of the collected scale data, and structural equation models were used to construct measurement models [50][51].

The variance of error for the aesthetic cognition was between 0.153 and 0.641, and the results were positive and significant. All standardized regression coefficients ranged from 0.61 to 0.9, with no coefficients above or overly close to 1.The stimated standard errors (SEs) of the variance of measurement errors ranged from 0.013 to 0.035, with no considerable SE observed. The variance of error for the aesthetic emotion was between 0.167 and 0.520, and the results were positive and significant. All standardized regression coefficients ranged from 0.632 to 0.890, with no coefficients above or overly close to 1.The stimated standard errors (SEs) of the variance of measurement errors ranged from 0.014 to 0.025, with no considerable SE observed.The variance of error for the aesthetic tendencies was between 0.195 and 0.505, and the results were positive and significant. All standardized regression coefficients ranged from 0.641 to 0.847, with no coefficients above or overly close to 1.The stimated standard errors (SEs) of the variance of measurement errors ranged from 0.018 to 0.029, with no considerable SE observed. The variance of error for the aesthetic behaviour was between 0.205 and 0.418, and the results were positive and significant. All standardized regression coefficients ranged from 0.679 to 0.845, with no coefficients above or overly close to 1.The stimated standard errors (SEs) of the variance of measurement errors ranged from 0.014 to 0.024, with no considerable SE observed. 

The results of the fitting tests for the four scales indicate that all values were within an acceptable range (presented in Table 1). Confirmatory factor analysis results indicated that the four scales had sufficient validity [52].

Insert Table 1 about here

Tests for convergent validity indicated that the standardized factor loadings of the aesthetic cognition scale, aesthetic emotion scale, aesthetic tendencies scale, and aesthetic behaviour scale were in the range of 0.61–0.9, 0.632–0.890, 0.641–0.847, and 0.679–0.845 respectively; these values were all greater than the acceptable criterion of 0.5 and were all significant [53][54].The combined reliability values for each dimension of each scale range from 0.781 to 0.915, all reaching the standard of being higher than 0.6 [55]. Average variant extraction (AVE) values ranged from 0.493 to 0.76, meeting the standard of being higher than 0.5 [55]. Therefore, the professional identity scale has sufficient convergent validity. The discriminant validity test indicated that the correlation coefficient of each dimension of the four scales was between 0.273 and 0.718, and a significant correlation was observed. The square root of AVE of each dimension of the scale was between 0.702 and 0.872, and the correlation coefficient value of each dimension was less than the square root of the AVE. And Cronbach's coefficients for the four scales ranged between 0.815 and 0.933. This indicates a certain correlation and a certain degree of discrimination between the latent variables. This also indicates that the four scales had sufficient discriminant validity [55]”.

The “gender” has been changed to the “sex”, and “through statistical analysis”has been deleted.

Furthermore，This article has been added with the limitations of the Part VI study.

Reviewer #2: This manuscript addresses a very interesting topic, not commonly addressed in higher education research and from a quantitative perspective. In this paper, the philosophical concept of aesthetic is discussed in the light of the relationship between psychological and behavioral students´characteristics. The study presents a complete literature review, however, sometimes, it lacks from the critical voices of the authors.

The methods section should describe the sample in a more detailed way (characteristics, access, selection criteria, informed consent, etc.). Results are presented in a clear and consistent manner. Summarizing, the manuscript presents a sound piece of scientific research with data supporting findings and sound conclusions. I recommend the paper for publication with minor corrections (Literature review, sample).

The author’s answer: We have revised the sentence in the article. The details as follows:

Written informed consent was given from all the participants prior to this study in accordance with the Declaration of Helsinki. Written informed consent was obtained from all the participants prior to this study.From 1 July 2022 to 8 July 2022, the researcher used Questionnaire Star software to distribute the questionnaires on the online platform, and 300 college students from five colleges and universities in Shaanxi Province, China, were selected as the official sample of the pretest for this study. The study distributed the official questionnaire to the participants three weeks after the pre-test questionnaire testing was completed and revised, the participants in the study were between the ages of 18 and 28 years old, and each participant received a cash prize for completing the questionnaire through the online platform. Written informed consent was obtained from all the participants prior to this study. 

Reviewer #3: Abstract:

It would be good to include some broader context into this abstract. Why is important to study students’ aesthetic behaviors? What is the purpose of the four scales? What is a daily aesthetic practice, and why would the reader want to know this? I can see from the intro that this is well developed in Chinese education, but it would be helpful for an international audience to have a bit more background.

Introduction:

• The introduction is quite long (2900 words), and reads a bit like a thesis. I think it can be streamlined and focused to bring the reader into the paper without having to explore each hypothesis in such detail.

• Beware of paragraphs that are too long, especially the first one, which can be difficult for the reader to follow.

• Early in the intro, it would be helpful to define aesthetic education.

Methods:

• It would help to know more about the participants. How many schools? What kinds of schools and what geographic area? What levels? How were they selected?

• There appears to be some results mixed into the methods section.

Discussion and Conclusion

• I liked these sections better. They were a bit more focused and brought things into a broader context. I liked the connection to specific recommendations for universities.

The author’s answer: We have revised the sentence in the article. The details as follows:

Firstly，Introduction section has been deleted (2372 words). 

Methods:The method section has been modified:

From 1 July 2022 to 8 July 2022, the researcher used Questionnaire Star software to distribute the questionnaires on the online platform, and 300 college students from five colleges and universities in Shaanxi Province, China, were selected as the official sample of the pretest for this study. The study distributed the official questionnaire to the participants three weeks after the pre-test questionnaire testing was completed and revised, the participants in the study were between the ages of 18 and 28 years old, and each participant received a cash prize for completing the questionnaire through the online platform. Written informed consent was obtained from all the participants prior to this study. 

A total of 1132 participants were received, with a total of 1060 valid questionnaires and a recovery rate of 93.6%. The source of formal samples are Shaanxi Normal University, Northwestern Polytechnical University, Xi’an University of Posts & Telecommunications, Northwest University, Northwest University of Political Science and Law, Xi`an International Studies University, Xidian University,Yulin Vocational and Technical College,Weinan Normal University,Yulin University,Xi’an University of Finance and Economics.The sample consisted of 589 (55.6%) males, 471 (44.4%) females, 422 (39.8%) art students, and 638 (60.2%) science students.

There is very little research on the everyday aesthetic behaviour of university students. This study uses structural equation modelling to conduct an empirical study. The study of the psycho-behavioural mechanisms of the aesthetic behaviours of university students is important for understanding the characteristics of aesthetic psychology and ensuring high-quality development of aesthetic education. 

Thank you very much for you attention and time. Look forward to hearing from you.

Yours sincerely,

Qiao Qiao, Yongzhi Jiang

Krirk University

8 Sep., 2023

---

## [Decision Letter · Decision Letter 1]

6 Oct 2023

PONE-D-23-22731R1The Influence of College Students' Aesthetic Cognitions on Aesthetic Behaviours: The Chain Mediation EffectPLOS ONE

Dear Dr. Jiang,

Thank you for submitting your manuscript to PLOS ONE. After careful consideration, we feel that it has merit but does not fully meet PLOS ONE’s publication criteria as it currently stands. Therefore, we invite you to submit a revised version of the manuscript that addresses the points raised during the review process.

We look forward to receiving your revised manuscript.

Kind regards,

Maja Vukadinovic

Academic Editor

PLOS ONE

Journal Requirements:

Reviewers' comments:

Reviewer's Responses to Questions

**Comments to the Author**

1. If the authors have adequately addressed your comments raised in a previous round of review and you feel that this manuscript is now acceptable for publication, you may indicate that here to bypass the “Comments to the Author” section, enter your conflict of interest statement in the “Confidential to Editor” section, and submit your "Accept" recommendation.

Reviewer #1: (No Response)

Reviewer #2: All comments have been addressed

2. Is the manuscript technically sound, and do the data support the conclusions?

Reviewer #1: Yes

Reviewer #2: Yes

3. Has the statistical analysis been performed appropriately and rigorously? 

Reviewer #1: Yes

Reviewer #2: Yes

4. Have the authors made all data underlying the findings in their manuscript fully available?

Reviewer #1: Yes

Reviewer #2: Yes

5. Is the manuscript presented in an intelligible fashion and written in standard English?

Reviewer #1: Yes

Reviewer #2: Yes

6. Review Comments to the Author

Reviewer #1: I thank the autors for addressing my previous commennts.

However, although the authors indicate that a section on limitations was added, I could not find (or see) this section!

A section on limitations should be inserted before the conclusions section.

Reviewer #2: The comments on methods section have been addressed and resolved. The comment on the literature review could be resolved by revising the way in which it was written, maybe to write it in a more critical way, by showing their voices too. However, it is only a recommendation because it has to do with the authors´writing style.

I recommend to accept the article.

7. PLOS authors have the option to publish the peer review history of their article (what does this mean?). If published, this will include your full peer review and any attached files.

Reviewer #1: **Yes: **Dr. Ali M. AL-Asadi

Reviewer #2: No

---

## [Author Response · Author response to Decision Letter 1]

7 Oct 2023

Dear reviewer,

 Thank you very much for your comments and professional advice. These opinions help to improve academic rigor of our article.Based on your suggestion and request, we have made corrected modifications on the revised manuscript, Meanwhile, the manuscript had be reviewed and edited by language servicices of AJE SEVIER. We hope that our work can be improved again. Furthermore, we would like to show the details as follows:

Reviewer #1: I thank the autors for addressing my previous commennts.

However, although the authors indicate that a section on limitations was added, I could not find (or see) this section!

A section on limitations should be inserted before the conclusions section.

Reviewer #2: The comments on methods section have been addressed and resolved. The comment on the literature review could be resolved by revising the way in which it was written, maybe to write it in a more critical way, by showing their voices too. However, it is only a recommendation because it has to do with the authors´writing style.

I recommend to accept the article.

The author’s answer: We have revised the sentence in the article. The details as follows:

“6 Research limitations 

6.1 Research Sample 

 In terms of sampling, limited by the researcher's time and ability, the sample of this study is only students of several universities in Shaanxi Province, China, and it is not possible to account for more Chinese college students in other provinces, which limits the interpretation of the results of the study and the inference. 

6.2 Research Variables 

 There are many factors that influence Chinese college students' aesthetic behavior, and this study is limited to two influences: aesthetic perception and aesthetic tendency. Other possible influencing factors, such as Chinese college students' personality traits, parents' income level, parents' attitudes toward art, the economic and cultural level of their cities, and the aesthetic environment, are not included in the variables of this study.

6.3 Research Method 

 This study uses a questionnaire survey to collect information about personal background variables, aesthetic cognition, aesthetic tendency and aesthetic behavior of Chinese college students in Shaanxi Province; however, whether subjects can truly express and reflect the real situation may have a biased influence on the correctness of the results of the study.”

Thank you very much for you attention and time. Look forward to hearing from you.

Yours sincerely,

Qiao Qiao, Yongzhi Jiang

Krirk University

7 Oct., 2023

---

## [Editor Report · Decision Letter 2]

10 Oct 2023

The Influence of College Students' Aesthetic Cognitions on Aesthetic Behaviours: The Chain Mediation Effect

PONE-D-23-22731R2

Dear Dr. Jiang,

We’re pleased to inform you that your manuscript has been judged scientifically suitable for publication and will be formally accepted for publication once it meets all outstanding technical requirements.

Kind regards,

Maja Vukadinovic

Academic Editor

PLOS ONE
---

## [Editor Report · Acceptance letter]

12 Oct 2023

PONE-D-23-22731R2 

The Influence of College Students' Aesthetic Cognitions on Aesthetic Behaviours: The Chain Mediation Effect 

Dear Dr. Jiang:

I'm pleased to inform you that your manuscript has been deemed suitable for publication in PLOS ONE. Congratulations! Your manuscript is now with our production department. 

Kind regards, 

on behalf of

Dr. Maja Vukadinovic 

Academic Editor

PLOS ONE